# Role of Berberine Thermosensitive Hydrogel in Periodontitis via PI3K/AKT Pathway In Vitro

**DOI:** 10.3390/ijms24076364

**Published:** 2023-03-28

**Authors:** Chang Wang, Chang Liu, Chen Liang, Xingyuan Qu, Xinying Zou, Siyu Du, Qian Zhang, Lei Wang

**Affiliations:** 1Department of Periodontology, Hospital of Stomatology, Jilin University, 1500 Tsinghua Road, Chaoyang District, Changchun 130021, China; wangchang20@mails.jlu.edu.cn (C.W.); liangchen21@mails.jlu.edu.cn (C.L.); quxy21@mails.jlu.edu.cn (X.Q.); siyud22@mails.jlu.edu.cn (S.D.); zhangqian20@mails.jlu.edu.cn (Q.Z.); 2Department of Prosthodontics, Hospital of Stomatology, Jilin University, Changchun 130021, China; jdliuchang@jlu.edu.cn; 3Department of Endodontics, Hospital of Stomatology, Jilin University, Changchun 130021, China; xyzou22@mails.jlu.edu.cn

**Keywords:** periodontitis, berberine, thermosensitive hydrogel, anti-inflammatory, osteogenesis

## Abstract

Periodontitis is a long-term inflammatory illness and a leading contributor to tooth loss in humans. Due to the influence of the anatomic parameters of teeth, such as root bifurcation lesions and the depth of the periodontal pocket, basic periodontal treatment on its own often does not completely obliterate flora microorganisms. As a consequence, topical medication has become a significant supplement in the treatment of chronic periodontitis. Berberine (BBR) has various pharmacological effects, such as hypoglycemic, antitumor, antiarrhythmic, anti-inflammatory, etc. The target of our project is to develop a safe and non-toxic carrier that can effectively release berberine, which can significantly reduce periodontal tissue inflammation, and to investigate whether berberine thermosensitive hydrogel can exert anti-inflammatory and osteogenic effects by modulating phosphatifylinositol-3-kinase/Protein Kinase B (PI3K/AKT) signaling pathway. Consequently, firstly berberine temperature-sensitive hydrogel was prepared, and its characterizations showed that the mixed solution gelated within 3 min under 37 °C with a hole diameter of 10–130 µm, and the accumulation of berberine release amounted to 89.99% at 21 days. CCK-8 and live-dead cell staining results indicated that this hydrogel was not biotoxic, and it is also presumed that the optimum concentration of berberine is 5 µM, which was selected for subsequent experiments. Real-time polymerase chain reaction (qRT-PCR) and Western blotting (WB)results demonstrated that inflammatory factors, as well as protein levels, were significantly reduced in the berberine-loaded hydrogel group, and LY294002 (PI3K inhibitor) could enhance this effect (*p* < 0.05). In the berberine-loaded hydrogel group, osteogenesis-related factor levels and protein profiles were visibly increased, along with an increase in alkaline phosphatase expression, which was inhibited by LY294002 (*p* < 0.05). Therefore, berberine thermosensitive hydrogel may be an effective treatment for periodontitis, and it may exert anti-inflammatory and osteogenic effects through the PI3K/AKT signaling pathway.

## 1. Introduction

Periodontitis is a type of infectious and immunological dental problem that leads to the foundation of the periodontal pocket, tooth movement, the resorption of the alveolar bone, and eventually tooth loss [1], and it is the leading cause of tooth loss worldwide [2]. It has been suggested that alterations to plaque bacteria can upset the balance between the hypo-gingival bacteria and the resident, contributing to injury of the periodontal tissue [3]. Accumulating evidence indicates that sustainable long-term inflammation decreases osteogenic differentiation and inactivates bone production [4,5]. For this reason, in addition to addressing the dangers of bacteria, the suppression of uncontrolled inflammation is critical to the treatment of periodontitis. The aim of treating periodontitis is to accomplish the reproduction of gum tissue [6]. Regeneration of the lead tissue is the earliest well-documented rejuvenation technique that uses the barrier film to facilitate the specific regeneration of periodontal membrane cells to repair periodontal defects. However, in most cases, complete regeneration is difficult to achieve [7]. Furthermore, pharmacological treatment includes antibiotics and nonsteroidal anti-inflammatory drugs (NSAIDs). However, there are some side effects to the use of these drugs. Therefore, the identification and development of secure, cost-effective, and cost-efficient drugs for the treatment of periodontitis are of great urgency [8].

In the last two decades, berberine has been discovered to have a variety of medicinal effects, such as hypoglycemic, antitumor, antiarrhythmic, anti-inflammatory, and immunomodulatory effects [9]. It has been shown to have good therapeutic effects, improving systemic and periodontal tissue as well as increasing alveolar bone density [10]. However, the efficacy of berberine is greatly limited on account of its low availability and non-targeting nature [11]. As a consequence of the anatomical irregularity of the periodontal pocket, an injection is a familiar form of administering medication for the therapy of periodontitis [12]. The carrier is inserted into the pocket of the periodontium in a non-invasive manner to allow for release in a continuous localized manner, an approach that would not only increase patient satisfaction but also boost the bioavailability of the drug [13]. In the meantime, it is to be noted that ongoing irrigation with gingival sulcus liquor will eliminate items from the periodontal pocket. For this reason, the optimal vehicle used for the therapy of periodontitis ought to possess excellent injective mobility and consistency after injection [14]. Often, in the clinical practice of periodontal therapy, injectable hydrogels offer the unique advantages of ease of preparation and use, low cost, and low toxicity, considering that increased drug concentrations and minimized adverse systemic reactions can be achieved simultaneously. They can be easily injected into irregularly shaped subgingival pockets and ensure uniform distribution throughout a specific site. These advantages make hydrogel significantly more competitive than other topical drug delivery systems [15]. Sodium alginate (SA) is a polysaccharide-based polymer [16]. Chitosan (CS) has been widely used in biomedicine due to its bioadhesive, non-toxic, biodegradable, antibacterial activity, antitumor and hemostatic properties, and is known, together with different biological sites of application, to form intrinsically biocompatible systems [17,18]. It was shown that hydrogel synthesized from these two substances has a porous construction, the ability to be used to carry cells and drugs, and at the same time, good mechanical strength [19]. In addition, the combination of chitosan and alginate molecules enhances the properties of chitosan to control drug release [20].

Berberine can promote periodontal tissue recovery and suppress inflammatory cytokine levels in experimental periodontitis in rats and decrease matrix metalloproteinase expression to prevent periodontitis-induced extracellular matrix decomposition [21]. There is still no clear mechanism of action and signal transduction pathway of berberine on inflammatory mediator expression and alveolar bone loss in periodontitis. Phosphatifylinositol-3-kinase/Protein Kinase B (PI3K/AKT) is a signaling pathway, a typical pathway that regulates inflammation and is frequently triggered during the progression of periodontitis. It has a major role in periodontal cell multiplication and apoptosis, the polarization of osteoblasts, and in the secretion of cytokines by influencing the activity of downstream effectors [22]. Phosphatidylinositol-3-kinase (PI3K) triggers AKT as well as downstream proteins, which participate in the regulation of cell survival, proliferation, and apoptosis [23]. AKT is a significantly lower target kinase in the PI3K signaling pathway [24]. Bone destruction is a natural consequence of the inflammatory process in periodontal tissues, and the development of periodontal disease can be influenced by a decrease in osteoblast activity, an increase in osteoclast activity, or both alterations [25]. Ghosh Choudhury et al. demonstrated that BMP-2 invokes osteoblast differentiation by means of activating the PI3k/AKT pathway [26]. LY294002 is a commonly used pharmacological inhibitor that has been widely used to inhibit PI3K signaling [27].

Lin et al. proved that the PI3K inhibitor, LY294002, can suppress the PI3K/AKT signaling axis, thereby preventing abnormal bone formation and reducing the degeneration of articular cartilage [28]. In this experiment, we evaluated the anti-inflammatory and osteogenic effects of a berberine thermosensitive hydrogel and showed that the berberine thermosensitive hydrogel might exert anti-inflammatory and osteogenic effects by regulating PI3K/AKT signaling pathway.

## 2. Results

### 2.1. Characterization of Hydrogel

We prepared the thermosensitive hydrogel through the physical mixing of CS, SA, and β-glycerophosphate (β-GP). The CS/β-GP/SA mixture was clear and transparent at room temperature and was converted from solution into the homogeneous gel in 3 min under 37 °C (Figure 1A). The scanning electron microscopy (SEM) results displayed that the hydrogel was multi-porous with a diameter of 10–130 μm (Figure 1B). The external spectra of the hydrogel and each component are illustrated in Figure 1C. Alginate has the typical bands owing to the symmetrical COO- and asymmetrical COO- groups at 1627 and 1417 cm^−1^. When complexed with chitosan, the bands become broader and shift slightly in width from 1627 to 1665 cm^−1^ and from 1417 to 1465 cm^−1^. The chitosan spectrum displays a C=O peak stretching of the amide at 1654 cm^−1^ and an N-H bending vibration of the non-acyl 2-amino glucose primary amine at 1562 cm^−1^. The drug release profile (Figure 1D) illustrated that in the first three days, there was an abrupt release of berberine, and the total aggregate percentage of drug release reached 60.07% on the 3rd day. The rate of release of the hydrogel decelerated and reached 84.19% on the 7th day. Gradual release with less drug release was observed after 7 days, and on the 21st day, the drug was released at 89.99%.

### 2.2. Cytotoxicity

In Figure 2A,B, there was no obvious comparison in cell viability between RAW264.7 cells incubated with various levels of berberine for 24 h and 48 h and the control group, except for the 10 μM group at 48 h. Under inverted fluorescence microscopy, the cell distribution was relatively uniform, and the number of live cells grew with increasing incubation time. The percentage of dead cells was less than 1% at both 24 h and 48 h under the low magnification (×10) field of view. In Figure 3A–C, when MC3T3-E1 cells were cultured with various levels of berberine for 1, 3, and 5 days, there was no obvious comparison in cell viability with the control group, except for the 10 μM group on the 5th day. In the live–dead cell staining assay, there was no difference between the groups. In addition, there was no difference between the dimethyl sulfoxide (DMSO) group and the control group, indicating that the dose of DMSO was not toxic.

### 2.3. Berberine Thermosensitive Hydrogel Mediates Anti-inflammatory Effect via PI3K/AKT Pathway

As shown in Figure 4A, after lipopolysaccharide (LPS) stimulation, the mRNA expressions in the berberine-loaded temperature-sensitive hydrogel group were inferior to the simple temperature-sensitive hydrogel group, and the LPS-stimulated group and the simple temperature-sensitive hydrogel group were remarkably superior to those in the blank group, and the expressions in the LY294002 group were inferior those in the berberine-loaded temperature-sensitive hydrogel group. In Figure 4B, the protein expressions of tumor necrosis factor-α (TNF-α), interleukin-6 (IL-6), and interleukin-1β (IL-1β) in the berberine-loaded temperature-sensitive hydrogel group were lower than that in the simple temperature-sensitive hydrogel group as well as the LPS-stimulated group, and the performance in the LY294002 group was lower than that in the berberine-loaded temperature-sensitive hydrogel group. As shown in Figure 4C, phosphorylated PI3K and AKT in the berberine-loaded temperature-sensitive hydrogel group were clearly lower than that in the temperature-sensitive hydrogel group and LPS-stimulated group alone, and the expression in the LY294002 group was lower than that in the berberine-loaded temperature-sensitive hydrogel group. As shown in Figure 4D, the expression of phosphorylated p65 in both the berberine-loaded temperature-sensitive hydrogel group was significantly lower than that in the simple temperature-sensitive hydrogel group and the LPS-stimulated group, and the LPS-stimulated and simple temperature-sensitive hydrogel groups were significantly higher than that in the blank group, and the performance in the LY294002 group was weaker than that in the berberine-loaded temperature-sensitive hydrogel group. The trend of phosphorylated IκBα was the same as phosphorylated p65.

### 2.4. Berberine Thermosensitive Hydrogel Mediates Osteogenesis Effect via PI3K/AKT Pathway

As shown in Figure 5A, after 7 d of osteogenesis induction, the expressions of alkaline phosphatase (ALP), type I collagen (COL-1), osteocalcin (OCN), and runt-related transcription factor 2 (Runx-2) in the berberine-loaded temperature-sensitive hydrogel group were higher than those in the simple temperature-sensitive hydrogel group and the empty group, the correspondence osteogenic gene expression was higher in the simple temperature-sensitive hydrogel group than that of the blank group, and the expression in the LY294002 group was less than that of the berberine-loaded temperature-sensitive hydrogel group. As shown in Figure 5B, all groups showed positive expression of alkaline phosphatase staining, and the degree of alkaline phosphatase staining in the berberine-loaded temperature-sensitive hydrogel group was higher than that in the blank group and higher than that in the simple temperature-sensitive hydrogel group, and the degree of alkaline phosphatase staining in the simple temperature-sensitive hydrogel group was deeper than that of the blank group, and the degree of alkaline phosphatase staining in the LY294002 group was lower than that of the berberine-loaded hydrogel group. As shown in Figure 5C, the degree of PI3K and AKT phosphorylation in the berberine-loaded temperature-sensitive hydrogel group was superior to that in the blank group and the simple temperature-sensitive hydrogel group, the degree of PI3K and AKT phosphorylation in the simple temperature-sensitive hydrogel group was deeper than that of the blank group, and the degree of PI3K and AKT phosphorylation in the LY294002 group was inferior to that in the berberine-loaded hydrogel group.

## 3. Discussion

Currently, oral health guidance, supragingival scaling, and subgingival scraping are the basic and most important effective methods to treat periodontal disease. Nevertheless, these methods alone often fail to eliminate plaque microorganisms due to dental anatomical conditions, including root bifurcation lesions and periodontal pocket depth [29]. Therefore, topical medication has become an important adjunct to the treatment of chronic periodontitis.

Hydrogels are three-dimensional polymer networks used as scaffolds for tissue regeneration or as delivery vehicles, which can absorb large amounts of water or biological fluids. The existence of high-level water content in hydrogels provides good biocompatibility, the ability to encapsulate hydrophilic drugs, and structural likeness to natural extracellular matrices or tissues. Composed by chemical and physical methods, such injectable hydrogels are prepared from various synthetic polymers whose degradation rates, products and times can be governed by the hydrophilic and hydrophobic balance in the copolymer or by the crosslink density [30]. Alginate contains mannose acid and gluonic acid, which are derived from seaweeds [31]. Alginate has appropriate properties such as enzymatic degradability and constituting an intrinsically biocompatible system [18,32]. The alginate bracket has several disadvantages, such as low mechanical strength and high degradation rates. To solve these problems, alginates are often blended with additional aggregates. Chitosan, which is derived from chitin, has excellent bio-properties, including bio-degradable, bio-safe and adhesion properties. Chitosan is licensed for biomedical applications, and yet, there are still significant concerns about its use in pharmaceuticals. The main regulatory issues with chitosan are attributed to its origin and properties. Its purity and other important issues are impurities, heavy metal contamination, proteins, microbial burden, and bacterial endotoxins. The level of purity of chitosan affects its physicochemical properties such as immunogenicity, solubility, and stability. Therefore, chitosan material for pharmaceutical use should have high purity [33]. In this experiment, the hydrogel was formed at 37°C. Scanning electron microscope images reveal the porous structure of the hydrogel, which ranges from 10-130 μm. A comparison of hydrogel constituents showed that the CS/β-GP/SA hydrogel was successfully synthesized. However, due to the interaction or superposition of groups, the amide peak shifts to 1665 cm^−1^ after complexation with alginate and the amino peak disappears (Figure 1B). One reason is the presence of more alginate within the scaffold, and another may be the presence of multiple interactions. These suggest that the protonated amino group has interacted with the carbonyl group. Hydrogel can carry small molecules such as drugs. The synthesis of this hydrogel can achieve a slow release of berberine in periodontal pockets, and the results showed that the cumulative release of berberine reached 89.99% in 21 days. Additionally, the gel time is about 3 min, which ensures that the hydrogel is not washed away by the gingival sulcus as much as possible.

Meanwhile, this experiment demonstrated that this hydrogel is not biotoxic according to both CCK-8 and the stain of living dead cells, showing that the hydrogel forms with the investigated biological test environment an intrinsically biocompatible system.

Periodontitis is a multi-factorial disease, with plaque being the main factor [34]. LPS originates from Gram-negative bacteria that accumulate at the surface of the teeth and is a major toxic factor contributing to the host vaccine reaction in periodontitis [35]. It has a crucial role in periodontitis by producing pro-inflammatory mediators, leading to periodontal tissue damage [36]. The qPCR and WB findings demonstrated great enhancement in the level of inflammatory factors in the LPS-stimulated group, indicating the successful establishment of an in vitro periodontitis model. Studies have shown that farro, quinoa, and other drugs suppressed the output of inflammatory factors or osteoclasts by controlling the PI3K/AKT signaling pathway [37]. In this experiment, inflammatory factors and PI3K/AKT-associated proteins of each group were measured with qPCR and WB techniques. The results displayed that the level was reduced in the berberine-loaded temperature-sensitive hydrogel group, indicating that the berberine-loaded temperature-sensitive hydrogel had anti-inflammatory effects. PI3K and AKT protein phosphorylation levels had a synergistic trend with periodontal inflammation levels, with reduced phosphorylation levels in the berberine-loaded temperature-sensitive hydrogel group, and LY294002 further reduced phosphorylation levels. This suggests to us that berberine may mediate periodontitis via suppression of the PI3K/AKT pathway.

Berberine exerts an influential role in suppressing inflammation by amplifying the NF-κB signaling pathway [38]. It was shown that induction of PI3K/AKT leads to signaling and translocation downstream of the NF-κB trans-transcription factor [39]. It was shown that the AKT-mediated NF-κB translocation was suppressed following LY294002 treatment [40]. Our experimental outcomes displayed that the level of phosphorylation of PI3K and AKT was reduced in the berberine thermosensitive hydrogel group, with a consequent reduction in the phosphorylation level of p65 and IκBα in the NF-κB pathway, and LY294002 further reduced phosphorylation levels. This suggests to us that berberine may mediate periodontitis through the PI3K/AKT/NF-κB pathway.

The dental alveolar skeleton is the most active of the human bone system, and its health has a direct impact on the health and function of the oromandibular system [41]. As a bone anabolic agent, berberine has not been developed as a clinical drug due to its side effects and low utilization [42]. The PI3K/AKT signaling pathway is critical in skeletal formation as well as bone remodeling [43]. In the ongoing investigation, we have studied the question of if berberine mediates the osteogenic differentiation that occurs in MC3T3-E1 cells along the PI3K/AKT signaling pathway. Previous studies have shown that RUNX2 is a major element in osteoblast differentiation; its increased expression indicates osteoinduction and osteoblast differentiation, OCN stimulates cell differentiation, and COL-1 is commonly known to be a marker of osteoblast polarization [44]. Within our work, the qPCR results showed that mRNA manifestations of RUNX2, ALP, OCN, and COL-1 were increased among the berberine-loaded temperature-sensitive hydrogel group, and LY294002 inhibited this effect. Alkaline phosphatase, a hallmark of primary osteogenic activity, is expressed in increased amounts in the process of osteogenic differentiation. Alkaline phosphatase catalyzes the breakdown of the phosphate ester to create ions that will be transferred to the outer cell matrix, thus contributing to the establishment of teeth and bone [45]. In what we studied, alkaline phosphatase activity became visibly intensified within the berberine thermosensitive hydrogel group, and LY294002 suppressed this effect. Western blotting (WB) indicated that increased levels of phosphorylation of PI3K and AKT were observed in the berberine-loaded temperature-sensitive hydrogel group, and this effect was inhibited by LY294002. With these results, berberine mediates the ossification of cells through the PI3K/AKT signaling pathway. In this experiment, it was found that the degree of alkaline phosphatase staining was higher in the simple temperature-sensitive hydrogel group than in the blank group, and various studies confirmed that such occurrence might be linked to the composition of the hydrogel. β-glycerophosphate sodium was found to promote the manifestation of various osteogenic-related genes, which is consistent with the qRT-PCR results of this experiment, which also increased alkaline phosphatase activity, collagen content, and calcification by increasing the phosphate content [46].

In conclusion, regarding the mode of administration of berberine, there are currently oral administration and the use of carriers. The hydrogel in this experiment can greatly improve the bioavailability of berberine compared with oral administration and achieve the purpose of slow release of the drug. Compared to other carriers such as nanoparticles as well as other hydrogels, thermosensitive hydrogel can be gelled at 37 °C, which matches body temperature and can be adapted to complex periodontal environments, and the fabrication process is easy and conducive to clinical trials. Moreover, it can be seen in the experimental results that the hydrogel has certain osteogenic properties, which can promote alveolar bone regeneration to a certain extent and contribute to the treatment of periodontitis while achieving the purpose of slow drug release. The present study provides some in vitro basis for the treatment of periodontitis with berberine. The mechanism related to the treatment of periodontitis is still not very clear; consequently, the experiment demonstrates great potential for research to be extended and the continued breaking of fresh ground concerning the treatment of periodontitis.

## 4. Materials and Methods

### 4.1. Synthesis of Hydrogel

Temperature-responsive chitosan hydrogel was prepared according to the previously reported method with a few modifications [47]. CS powder (Aladdin, Shanghai, China) was dissolved in a 0.1 mol/L ethanoic acid aqueous solution and configured into a 2% (*w*/*v*) chitosan solution. Sodium alginate powder (Macklin, Shanghai, China) was mixed with De-ionized water to form 0.6% (*w*/*v*) aqueous sodium alginate solution. Next, β-GP (Macklin, Shanghai, China) was dispersed in di-ionized water to make a β-GP (*w*/*v*) solution with a quality content of 56%. CS, β-GP and SA solutions were equipped with a *v/v* ratio of 1:1:2. The β-GP solution was initially taken drop by drop into the CS solution at 4 °C, and after mixing well, the aqueous SA solution was incorporated drop by drop under magnetic mixing until the solution was well integrated. The pH was tuned to neutral by using 0.1 mol/L sodium hydroxide (NaOH) solution to attain a temperature-sensitive hydrogel of CS/β-GP/SA. For the next step, 2 mL of the hydrogel was moved to a sterile ampoule, closed and stored in a temperature-controlled under 37 °C. Record the duration of the gel using the test tube inversion method.

### 4.2. Characterizations of Hydrogel

#### 4.2.1. Morphological Characteristics

CS/β-GP/SA hydrogel was pre-cooled under −80 °C for 24 h before freeze-drying. The microstructure was further observed using an SEM (Carl Zeiss Jena, Oberkochen, Germany) [48].

#### 4.2.2. FTIR

One mg of lyophilized CS/β-GP/SA hydrogel was combined with the suitable dosage of Kbr dust and pressed into tablets. The thermosensitive hydrogel was analyzed for the architecture and the organic functional group composition in the range of 500–3000 cm^−1^ using FTIR (Nicolet Instrument Corporation, Madison, WI, USA) [49].

#### 4.2.3. Drug Release Rate

Berberine/CS/β-GP/SA was obtained by dropping berberine (Yuanye, Shanghai, China) dissolved with DMSO (Sigma, Munich, Germany) into CS/β-GP/SA aqueous solution under a sterile environment and mixed well. Berberine/CS/β-GP/SA mixture pre-gelatinization was loaded into 6-well plates (2.5 mL per well) and placed in a 37 °C incubator to fully incubate the gel. After being incubated, 8 mL of simulated body fluid (SBF) was incorporated into every well and was placed at 37 °C for further incubation. At 2 h, 4 h, 8 h, 12 h, 1, 3, 7, 14, and 21 days of incubation, 3 mL of supernatant was gathered from each well and replenished with the same amount of SBF solution into the wells. The optical density (OD) values of the collected supernatants were measured using an enzyme marker, and the concentrations of berberine in the supernatants at each time point were quantified. The accumulative release rate was then calculated on the basis of formulation: Fi = (3∑C_i − 1_ + 8 Ci)/(dosage × drug content). Fi is the system’s cumulative release of the sample, Ci is the concentration of drug release for the ith sample, 3 is the amount of each sample (unit: mL), and 8 is the gross amount of the release system (unit: mL), thereby plotting the cumulative release curve [48].

### 4.3. Cytotoxicity Assay

Raw 264.7 and MC3T3-E1 cells were raised in DMEM with supplementation of 10% FBS and 1% penicillin–streptomycin, growing stably in a humid environment at 5% CO_2_ and 37 °C. The cells were inoculated in 96-well plates (5 × 10^3^ cells/well) and cultured for 24 h. Next, replace the fresh medium that contained different concentrations of berberine thermosensitive hydrogel, which is made by a medium obtained by co-culturing temperature-sensitive hydrogel with or without berberine with DMEM for 24 h. Raw 264.7 was incubated for 24 h and 48 h; MC3T3-E1 was incubated for 1, 3 and 5 days. Add 10 μL CCK-8 reagent to every well and incubate for 2 h under 37 °C. Finally, the OD of the medium would be calculated under 450 nm by an enzyme marker (Lei Du Life Science and Technology Co., Shenzhen, China) and the vitality of the cells was established as follows: cell viability (%) = (test OD − blank OD)/(control OD − blank OD) × 100%. Used for live/dead cell staining, cells were inoculated in 6-well plates (2 × 10^5^ cells/well) and incubated for 24 h. The fluid changes and incubation times were the same as for the CCK-8 method, and the staining was incubated with live/dead cell dye (Beyotime, Shanghai, China), live cells were colored green by fluorescently excited calcein AM and the dead cells were colored red by propidium iodide (PI) and observed under fluorescence microscopy to detect hydrogel toxicity [48].

### 4.4. Anti-Inflammatory Effect

#### 4.4.1. qRT-PCR

Raw264.7 cells were spread in six-well plates at 2 × 10^5^ per well, and after attachment, an inhibitor (LY294002, MCE) was added to the inhibitor group, and after half an hour, berberine was added for pretreatment, and after 3 h, LPS was added and stimulated for 24 h. The mRNA of the Raw264.7 cells was assessed with a TRIeasy™ Total RNA Extraction Reagent (YEASEN, Shanghai, China), and the cDNA was obtained with Reverse Transcription Kit (YEASEN, Shanghai, China). qRT-PCR was conducted on a CFX96 qRT-PCR detection system (Bio-Rad, Hercules, CA, USA) to obtain the cDNA. The mRNA levels of IL-6, IL-1β, and TNF-α were measured. The mRNA expression levels were assayed using the 2^−ΔΔCt^ comparison method with an internal control gene (β-actin) as the standard. The primer sequences are summarized in Table 1 [50].

#### 4.4.2. Western Blotting

For the cell culture as seen in Section 4.4.1, Raw264.7 cells were gathered before being lysed by using radioimmunoprecipitation (RIPA, Beyotime, Shanghai, China), which contains inhibitors of protease. The cells were isolated by centrifugation under 4 °C for 30 min, and the upper liquid was gathered. Protein measurements should be made using the BCA (Beyotime, Shanghai, China), and the volume of supernatant should be quantified at the same time. Then, depending on the supernatant volume, the upper sample buffer (Beyotime, Shanghai, China) was added, and the protein sample obtained was boiled at 100 °C for 5 min and kept at −20 °C. The gross protein (20 μg) was loaded onto dodecyl sulfate–polyacrylamide gel, and electrophoresis was performed at 80 V for 30 min, followed by switching the voltage to 120 V before the labeled dye arrived roughly 1 cm from the underside of the gel. Transfer the protein onto a nitrocellulose membrane using a wet method with 300 mA in 90 min. Wash the membrane, block it with Fast Closure Buffer (Beyotime, Shanghai, China) for 15 min at 4 °C, and then incubate it with the first antibody overnight. Primary antibodies included rabbit anti-IL-6 (1:500, Wanleibio, Shenyang, China), TNF-α (1:1000, Abcam, Cambridge, UK), IL-1β (1:1000, Abcam, Cambridge, UK), IκBα (1:500, Wanleibio, Shenyang, China), p-IκBα (1:500, Wanleibio, Shenyang, China), p65 (1:1000, Abcam, Cambridge, UK), p-p65 (1:500, Wanleibio, Shenyang, China), AKT (1:500, Wanleibio, Shenyang, China), p-AKT (1:500, Wanleibio, Shenyang, China), PI3K (1:500, ProteinTech, Wuhan, China), p-PI3K (1:500, ProteinTech, Wuhan, China), and mouse antibody β-actin (1:1000, ProteinTech, Wuhan, China). Lastly, the films were hatched using HRP-coupled secondary antibodies on an oscillator with 1 h incubation. The secondary antibodies consisted of HRP-tagged goat anti-rabbit IgG (Beyotime, Shanghai, China) as well as goat anti-mouse IgG (Beyotime, Shanghai, China). To visualize protein bands by enhancing the chemiluminescent reagent (NCM Biotech, Suzhou, China). The strip intensity was calculated using ImageJ software [48].

### 4.5. Osteogenesis

#### 4.5.1. qRT-PCR

Similar to Section 4.4.1, except that Raw264.7 cells are replaced with MC3T3-E1 cells, without LPS, change the osteogenic induction fluid every three days. After seven days of incubation, the mRNA levels ALP, COL-1, OCN, and Runx2 were measured. The primer sequences are summarized in Table 2.

#### 4.5.2. ALP Staining

MC3T3-E1 cells were inoculated in 6-well plates (5 × 10^4^ cells/well) and cultured for 24 h. The osteogenic induction solution containing 10 mM sodium β-GP, 5 mg/mL vitamin C, and 10 mM dexamethasone was changed every three days. The medium required for the preparation of osteogenic induction solution was obtained from hydrogels with or without berberine co-cultured with DMEM for 1 day. After seven days of incubation, the cells were stationary in 4% paraformaldehyde for 30 min under room temperature and then stained according to the instructions of the alkaline phosphatase color development kit (Beyotime) and were observed by microscopy.

#### 4.5.3. Western Blotting

As with Section 4.4.2, except that Raw 264.7 cells are replaced with MC3T3-E1 cells. Protein collection after seven days of incubation. Primary antibodies included rabbit anti-AKT, p-AKT, PI3K, p-PI3K, and mouse antibody β-actin.

### 4.6. Statistical Analysis

It was duplicated at least three times for all experiments in triplicate. For the statistical analysis, Graphpad prism 8 (USA) was used. In all cases, the data were presented as mean ± standard deviation. Comparisons of multiple sets were made by one-way analysis of variance, and *p* < 0.05 was deemed to be scientifically relevant.

## 5. Conclusions

Berberine CS/β-GP/SA thermal hydrogel has good biocompatibility and can achieve prolonged and constant drug release, which can effectively suppress the performance of the inflammatory agents, promote the expression of osteogenic factors, and consistently and effectively generate anti-inflammatory and osteogenic benefits in periodontal tissues. Therefore, berberine CS/β-GP/SA thermosensitive hydrogel is an innovative and effective new approach for the treatment of chronic periodontitis.

## Figures and Tables

**Figure 1 ijms-24-06364-f001:**
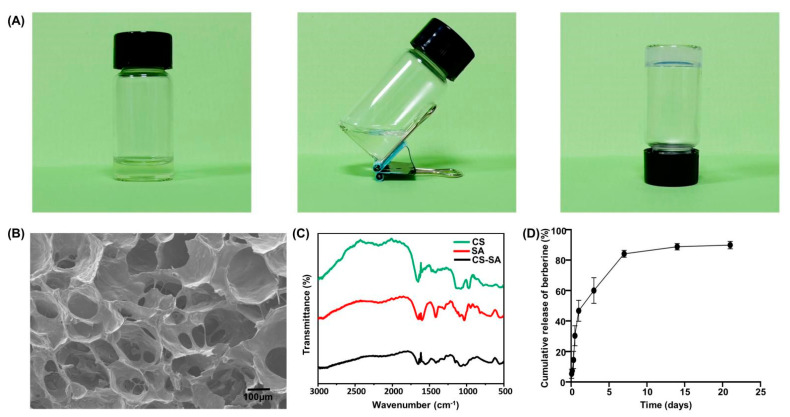
Characterizations of the thermosensitive hydrogel. (**A**) Image of CS/β–GP/SA mixed solution transformed into hydrogel after 3 min at 37 °C. (**B**) SEM image of the thermosensitive hydrogel. (**C**) Fourier transform infrared spectrometry (FTIR) spectra for thermosensitive hydrogel. (**D**) The release percentage curve of berberine from the thermosensitive hydrogel.

**Figure 2 ijms-24-06364-f002:**
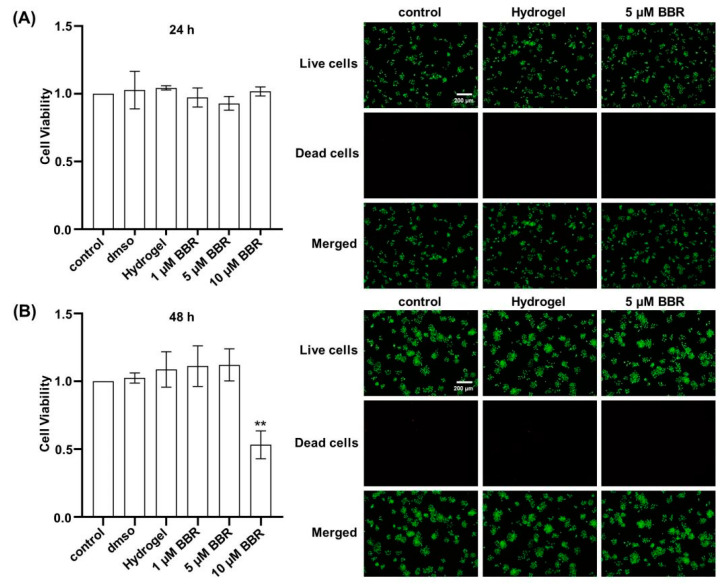
Cytotoxicity of berberine-loaded CS/β-GP/SA thermosensitive hydrogel in RAW 264.7 cells. Viability of RAW 264.7 cells cultured with berberine-loaded CS/β-GP/SA thermosensitive hydrogel at different concentrations (1–10 μM) at 24 h (**A**) and 48 h (**B**). Green represents living cells, and red represents dead cells. **: *p* < 0.01.

**Figure 3 ijms-24-06364-f003:**
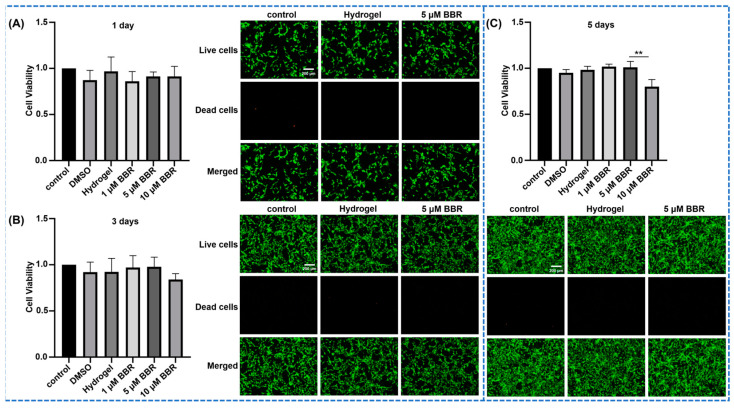
Cytotoxicity of berberine-loaded CS/β-GP/SA thermosensitive hydrogel in MC3T3-E1 cells. Viability of MC3T3-E1 cells cultured with berberine-loaded CS/β-GP/SA thermosensitive hydrogel at different concentrations (1–10 μM) at 1 Day (**A**), 3 Days (**B**) and 5 Days (**C**). Green represents living cells, and red represents dead cells. **: *p* < 0.01.

**Figure 4 ijms-24-06364-f004:**
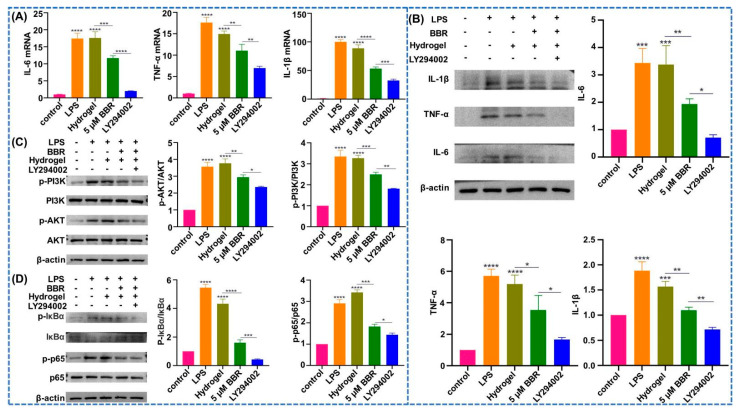
Anti-inflammatory effect of berberine thermosensitive hydrogel. (**A**) Change of IL–1β, IL–6, and TNF–α mRNA expression; (**B**) Change of IL–1β, IL–6, and TNF–α protein expression; (**C**) Protein changes in the PI3K/AKT pathway; (**D**) Protein changes in the nuclear factor kappa−B (NF−κB) pathway. *: *p* < 0.05, **: *p* < 0.01, ***: *p* < 0.001, ****: *p* < 0.0001.

**Figure 5 ijms-24-06364-f005:**
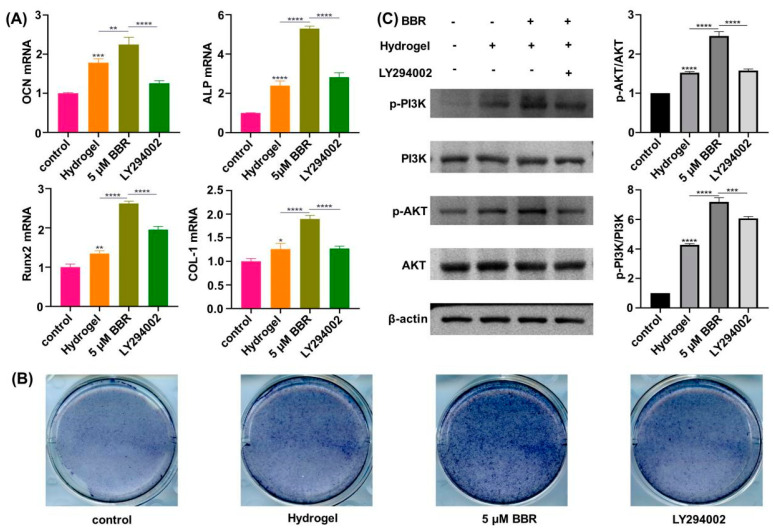
Osteogenesis effect of berberine thermosensitive hydrogel. (**A**) Change of ALP, COL−1, OCN, Runx−2 mRNA expression; (**B**) Alkaline phosphatase staining results; (**C**) Protein changes in the PI3K/AKT pathway. *: *p* < 0.05, **: *p* < 0.01, ***: *p* < 0.001, ****: *p* < 0.0001.

**Table 1 ijms-24-06364-t001:** Primer sequence for qRT-PCR in anti-inflammatory study.

Gene	Sequence of Primers (5′→3′)
β-actin	F: GGAGATTACTGCCCTGGCTCCTA R: GACTCATCGTACTCCTGCTTGCTG
IL-6	F: AAGCCAGAGCTGCAGGATGAGTA R: TGTCCTGCAGCCACTGGTTC
TNF-α	F: TTCCAATGGGCTTTCGGAAC R: AGACATCTTCAGCAGCCTTGTGAG
IL-1β	F: CCCTGAACTCAACTGTGAAATAGCA R: CCCAAGTCAAGGGCTTGGAA

**Table 2 ijms-24-06364-t002:** Primer sequence for qRT-PCR in osteogenesis study.

Gene	Sequence of Primers (5′→3′)
β-actin	F: CATCCGTAAAGACCTCTATGCCAAC R: ATGGAGCCACCGATCCACA
ALP	F: GCAGTATGAATTGAATCGGAACAC R: ATGGCCTGGTCCATCTCCAC
COL-1	F: GACATGTTCAGCTTTGTGGACCTC R: GGGACCCTTAGGCCATTGTGTA
OCN	F: AGCAGCTTGGCCCAGACCTA R: TAGCGCCGGAGTCTGTTCACTAC
Runx-2	F: TGCAAGCAGTATTTACAACAGAGG R: GGCTCACGTCGCTCATCTT

## Data Availability

The data in this study are accessible from the corresponding authors upon request.

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
