# Peer review of "Role of Berberine Thermosensitive Hydrogel in Periodontitis via PI3K/AKT Pathway In Vitro"

_ijms, 2023, doi:10.3390/ijms24076364_

Round 1

Reviewer 1 Report

This paper entitled "Role of Berberine Thermosensitive Hydrogel in Periodontitis via PI3K/AKT Pathway in Vitro" by Wang et al. developed a berberine thermosensitive hydrogel.

This study developed a berberine thermosensitive hydrogel. However, berberine is well known as an anti-inflammatory chemical, and its regulation of the PI3K/AKT pathway has also been reported. There are different types of delivery methods available, and the authors may want to clarify the advantage and innovation of their hydrogel.

The overall quality of the work is good. The manuscript is well prepared, and the data is convincing. However, some concerns need to be addressed as follows:

Major Concerns:

1 All the figures are too small to be seen.

2 In Figure 1d, it appears that most of the berberine is released within hours. The authors should provide finer data points in hours.

3 The advantage and innovation of the hydrogel compared to other berberine delivery methods should be further discussed.

Minor Issues:

1 The authors may want to define the CS/b-GP/SA before using the abbreviations, as the results section precedes the materials and methods section.

2 The hydrogel may not be clear due to the green background.

3 The scales in Figure 1b could be clearer.

Reviewer 2 Report

The paper presents a safe and non-toxic carrier that can effectively release berberine, which can significantly reduce periodontal tissue inflammation, and to investigate whether berberine thermosensitive Hydrogel hydrogel can exert anti-inflammatory and osteogenic effects by modulating PI3K/AKT signaling pathway.  It is a topic of interest in researchers in the related areas, but the paper needs very significant improvement before acceptance for publication.  My detailed comments are as follows:

1.  Where is the basis of the experimental plan and detection method, and the literature supports not enough.

2.  In Figure 2A and Figure 2B, What are the setting conditions of control group and dmso group?  What are the possible reasons for the trend as shown in the figure? 

3.  The chapter arrangement is not reasonable enough.

4.  The titles of fig 2 and fig 3 in part 2.2 are the same.

5.  Introduction section is not concise enough, maybe you should improve it.

6.  The release percentage curve of IL-1ra from CS/β-GP/SA hydrogel shows that it has reached equilibrium around the 7th day.  What is the sustained release mechanism of hydrogel?

7.  The format of references should be consistent.

Round 2

Reviewer 1 Report

I appreciate the author's effort in responding to my feedback. While I am generally satisfied with their response, I would like to request that they provide more detailed information about the advantages and innovations of their hydrogel compared to other berberine delivery methods. I have come across several reports on oral delivery, various berberine nanoparticles, and different berberine hydrogels. Therefore, I would suggest that the authors include an up-to-date comparison between these methods to highlight the advantages of their innovation. Such a comparison would be beneficial in fully understanding the unique features of their hydrogel and how it stands out from the alternatives.

Round 3

Reviewer 1 Report

I am satisfied with the revisions and would like to endorse it for publication.
